# Childhood Hypophosphatasia Associated with a Novel Biallelic *ALPL* Variant at the TNSALP Dimer Interface

**DOI:** 10.3390/ijms24010282

**Published:** 2022-12-23

**Authors:** Luciane Martins, Luis Gustavo F. Lessa, Taccyanna M. Ali, Monize Lazar, Chong A. Kim, Kamila R. Kantovitz, Mauro P. Santamaria, Cássia F. Araújo, Carolina J. Ramos, Brian L. Foster, José Francisco S. Franco, Débora Bertola, Francisco H. Nociti

**Affiliations:** 1Department of Community Medicine and Epidemiology, CHS National Cancer Control Center, Haifa 3436212, Israel; 2Department of Research, São Leopoldo Mandic School of Dentistry and Research Center, Campinas, São Paulo 13045-755, Brazil; 3Centro de Estudos do Genoma Humano e Células-Tronco, Instituto de Biociências, Universidade de São Paulo, São Paulo 05508-090, Brazil; 4Clinical Genetics, Instituto da Criança do Hospital das Clínicas da Faculdade de Medicina da Universidade de São Paulo, São Paulo 05403-000, Brazil; 5Department of Biomaterials, São Leopoldo Mandic School of Dentistry and Research Center, Campinas, São Paulo 13045-755, Brazil; 6Center for Oral Health Research, University of Kentucky, College of Dentistry, Lexington, KY 40536, USA; 7Department of Diagnosis and Surgery, Institute of Science and Technology, São Paulo State University (UNESP), São José dos Campos, São Paulo 12245-000, Brazil; 8Division of Biosciences, College of Dentistry, The Ohio State University, Columbus, OH 43210, USA; 9Center of Research and Molecular Diagnosis of Genetic Diseases, Department of Biophysics, UNIFESP and Biotecnologic Centre, Energy and Nuclear Research Institute (IPEN), Universidade de São Paulo, São Paulo 05508-000, Brazil

**Keywords:** alkaline phosphatase, premature tooth loss, hypophosphatasia, genotype–phenotype association, 3D modeling

## Abstract

The goal of this study was to perform a clinical and molecular investigation in an eight-year-old female child diagnosed with hypophosphatasia (HPP). The proband and her family were evaluated by medical and dental histories, biochemical analyses, radiographic imaging, and genetic analysis of the tissue-nonspecific alkaline phosphatase (*ALPL*) gene. A bioinformatic analysis was performed to predict the structural and functional impact of the point mutations in the tissue-nonspecific alkaline phosphatase (TNSALP) molecule and to define their potential contribution to the phenotype. We identified a novel combination of heterozygous *ALPL* missense variants in the proband, p.Ala33Val and p.Asn47His, compatible with an autosomal recessive mode of inheritance and resulting in skeletal and dental phenotypes. Computational modeling showed that the affected Asn47 residue is located in the coil structure close to the N-terminal α-helix, whereas the affected Ala33 residue is localized in the N-terminal α-helix. Both affected residues are located close to the homodimer interface, suggesting they may impair TNSALP dimer formation and stability. Clinical and biochemical follow-up revealed improvements after six years of ERT. Reporting this novel combination of *ALPL* variants in childhood HPP provides new insights into genotype–phenotype associations for HPP and specific sites within the TNSALP molecule potentially related to a childhood-onset HPP and skeletal and dental manifestations. Beneficial effects of ERT are implicated in skeletal and dental tissues.

## 1. Introduction

Hypophosphatasia (HPP; OMIM# 146300, 241500, 241510) is a rare hereditary disorder caused by loss-of-function mutations in the *ALPL* gene (MIM*171760) that encodes the tissue-nonspecific isozyme of alkaline phosphatase (TNSALP; UniprotKB# P05186). This inborn error-of-metabolism is characterized by defective skeletal and dental mineralization and other less well-understood complications resulting from deficient TNSALP activity and accumulation of its substrates [1]. Individuals with HPP have markedly heterogeneous clinical manifestations and are classified with one of the clinical subtypes of HPP based on the age of onset and severity. Biallelic variants in *ALPL* are associated with more severe phenotypes, whereas heterozygous variants in this gene are compatible with an autosomal dominant pattern of inheritance and milder phenotypes. Manifestations of HPP include defective bone mineralization with bone deformities, osteomalacia and osteopenia, spontaneous fractures, short stature/rickets, premature craniosynostosis, arthropathy, premature loss of deciduous teeth, enamel and dentin defects, and seizures [1].

The clinical suspicion of HPP is confirmed by low levels of serum alkaline phosphatase activity (ALP) and genetic screening of the *ALPL* gene [1]. Over 400 distinct *ALPL* variants have been described in association with HPP, the majority being missense variants (https://alplmutationdatabase.jku.at/table/, accessed on 8 December 2022). Several missense variants have been demonstrated to affect residues important for protein stability and function [2,3]. Here, we present a case of a female child diagnosed with a novel combination of *ALPL* compound heterozygous missense variants associated with childhood HPP, which presented significant clinical improvements following asfotase alfa enzyme replacement therapy (ERT) for six years.

## 2. Methods

### 2.1. Clinical Assessment

The study was conducted in compliance with the recommendations of the World Medical Association Declaration of Helsinki, Ethical Principles for Medical Research Involving Human Subjects (IRB # CAPpesq protocol # 0558/10). The proband is an eight-year-old Caucasian female with childhood-onset HPP. Other family members, including her mother, father, and sister, were clinically and genetically evaluated in order to identify inheritance and genotype–phenotype association patterns. Clinical growth charts to evaluate normal growth range for children older than two years and determine percentiles and Z-scores for the proband were referenced at the Centers for Disease Control (CDC) website (http://www.cdc.gov/growthcharts, accessed on 8 December 2022).

### 2.2. Genetic Analysis

Initially, a chromosomal microarray analysis was performed, followed by a targeted next-generation sequencing (NGS) customized panel, including 130 genes responsible for Mendelian disorders presenting mainly proportionate syndromic short stature and skeletal disorders (Illumina, Inc; San Diego, CA, USA).

### 2.3. Residue Conservation Analysis

Multiple species amino acid sequence alignment for TNSALP was performed using Clustal Omega (http://www.ebi.ac.uk/Tools/msa/clustalo/, accessed on 8 December 2022). Orthologous TNSALP protein sequences from 15 vertebrate species were used for the multiple sequence alignment, including: human (ALPL: NP_000469.3; ALPP: NP_001623.3; ALPI: NP_001622.2; ALPPL: NP_112603.2), chimpanzee (XP_016811317.1), monkey (NP_001253798.1), mouse (NP_031457.2), rat (NP_037191.2), dolphin (XP_033697674.1), dog (NP_001184066.1), cat (NP_001036028.1), pig (XP_020953341.1), sheep (XP_012008214.3), horse (XP_023491337.1), cow (NP_789828.2), chicken (NP_990691.1), frog (XP_031762314.1), and zebrafish (NP_957301.2) sequences.

### 2.4. Mutation Pathogenicity Prediction

PolyPhen and Mutation Taster predictive in silico tools were used to perform the mutation pathogenicity prediction and predict the impact of p.Asn47His in the protein function and structure, based on sequence homology and physical proprieties of amino acids [4,5].

### 2.5. Homology Modeling for TNSALP

Three-dimensional (3D) models for native and mutant TNSALP proteins were built based on the crystal structure of human placental alkaline phosphatase (PDB ID: 1EW2) [6] (Swiss-Model, Expasy web server). The models were aligned, visualized, and analyzed using the open-source PyMOL software (PyMOL Molecular Graphics System, Version 1.7.4, Schrödinger, LLC), and intra and intermolecular interactions were visualized in the Swiss Pdb-Viewer software.

## 3. Results

### 3.1. Medical and Dental History

The proband is an eight-year-old girl, the second child of non-consanguineous parents. There is no history of skeletal anomalies in her family, including in her older sister. The proband was evaluated at 11 months of age because of motor developmental delay and failure to thrive. A physical exam showed a weight of 6800 g (Z-score, −2.07), length of 65 cm (Z-score, −2.48), and occipital frontal circumference (OFC) of 46.5 cm (Z-score, 1.42). The dolichocephalic skull had no facial dysmorphisms, and the patient had a pectus carinatum with a bell-shaped thorax. The skeletal survey disclosed abnormal, heterogeneous trabecular bone in the ilium bone of the pelvis, proximal femur, and the long bones at the knees, with radiolucent areas in the proximal metaphysis of the fibula and in the central metaphysis and epiphysis of the left distal femur (Figure 1A,B). The proband showed enlargement of the anterior costochondral junction (Figure 1C). No abnormalities were observed for the long bones of the upper limbs (Figure 1D). A 3D computed tomography scan of the skull disclosed premature fusion of the sagittal and metopic sutures and a partial fusion of the lambdoid sutures, requiring surgical correction at 23 months of age (Figure 1E).

The pregnancy was unremarkable, and the proband was born at term (40 weeks) by Cesarean section, with a birth weight of 3335 g (Z-score, 0.24), length of 48 cm (Z-score, −0.65), occipital frontal circumference (OFC) of 38 cm (Z-score, 1.77), and Apgar scores of 9 and 10. In the first months of life, she developed recurrent vomiting, with diagnoses of gastroesophageal reflux and allergy to cow’s milk protein, which improved with the introduction of a different formula and anti-reflux medication. Despite amelioration of vomiting, the proband evolved with failure to thrive, along with motor delay and irritability. She was able to sit unsupported at 8 months of age and walked independently at 23 months, but with a waddling gait.

Genetic analysis revealed biallelic variants in the *ALPL*, the gene associated with HPP. The nature and inheritance of the variants are described in more detail in the next section. At the age of 1 year and 9 months, biochemical analysis showed borderline high (but within normal range) levels of serum calcium (11.0 mg/dL; reference values (RV): 9–11 mg/dL) and phosphorus (6.3 mg/dL; RV: 3.4–6.2 mg/dL), reduced ALP levels (34 U/L; RV: 70–350 U/L), and elevated levels of vitamin B6 (250 μg/L; RV: 5.2–34.1 μg/L), supporting the diagnosis of HPP.

ERT with asfotase alfa was administered subcutaneously to the proband (2 mg/kg; 3×/week) starting at the age of 2 years and 10 months. ERT resulted in improvement in her pain and anthropometric measurements, as well as her gait and motor skills, though these lagged compared to the motor achievements of her peers. In addition, bone mineral density (BMD) (total body and lumbar spine (L1–L4)) was periodically obtained after the first year of ERT by dual-energy X-ray absorptiometry (DXA), and data analysis showed BMD Z-scores within the normal lower range for both total body and lumbar spine, ranging from −3.4 (at the age of 3 years) to −2.9 (at the age of 7 years). During the first six years of ERT, the proband experienced some mild skin reactions such as redness, swelling, and thickening (Figure 1F–H). More recently, she developed lipodystrophy at injection sites, mainly in her thighs and arms (Figure 1I). The radiolucent anomaly detected at the proximal right fibula, and a transverse fracture in the proximal fibula diaphysis, showed substantial resolution after initiating ERT, based on follow-up radiographs at 3 years and 5 months and at 4 years of age (Figure 1J). No recurrent bone fractures occurred after initiation of ERT, but the proband has complained of occasional pain in the lower limbs. The Z-scores of her weight and height before the ERT (at 2 years and 3 months) were −2.84 and −2.07, respectively. These improved to −1.50 and −1.47, respectively, at the age of 3 years and 5 months, after 7 months of ERT. Currently, her weight is 28 kg and height is 1.20 m (Z-scores, 0.04 and −1.86, respectively).

A detailed dental assessment was not performed until the proband was eight years of age. Oral examination revealed areas of mild enamel hypoplasia and an accelerated exfoliation of deciduous teeth, including canines and molars (Figure 1K–N). The proband exhibited good oral hygiene with a plaque index of 26% and bleeding index of 28%, no signs of gingival recession, and no pathological swellings. Periodontal probing showed normal attachment levels and no sign of periodontal pocketing due to alveolar bone loss (data not shown). Upon collection of medical/dental history information, the proband’s mother recalled that eruption of her first teeth was delayed, occurring at 16 months of age. Two deciduous mandibular incisors were prematurely lost with intact roots at the age of three years and six months. In addition, the proband’s maternal great grandmother reportedly lost permanent teeth at a young age (30 s).

### 3.2. Genetic and Pedigree Analysis

Genetic analysis revealed the proband is a compound heterozygous carrier of two *ALPL* missense variants, both located in exon 3. The variant inherited from the paternal side is a heterozygous substitution A → C at position 139-nt, leading to the substitution of asparagine for histidine 47 (p.Asn47His). The maternally inherited variant is a heterozygous substitution C → T at position 98-nt, leading to the substitution of alanine for valine at position 33 (p.Ala33Val) (Figure 2A, B). Both variants were classified as likely pathogenic (American College of Medical Genetics and Genomics (ACMG) criteria). The latter is also present in the proband’s sister. Pedigree analysis, therefore, indicated an autosomal recessive inheritance pattern (in compound heterozygozity—p.[Ala33Val];[Asn47His]) because both parents and her sister were asymptomatic heterozygous carriers of only one *ALPL* variant and exhibited normal serum ALP activity. Based on the proband’s medical and dental history, pedigree evaluation, low serum ALP levels, elevated levels of pyridoxine, and identification of *ALPL* mutations, a diagnosis of childhood HPP was supported.

### 3.3. Bioinformatic Analysis of ALPL Mutations

Both p.Ala33Val and p.Asn47His variants were predicted as disease-causing with a probability of 0.99 by Mutation Taster and predicted as possibly damaging, with a score of 0.863 and 0.866, respectively (sensitivity: 0.72; specificity: 0.89), using the HumVar classification models of Polyphen2. Sequence alignments revealed that both Ala33 and Asn47 are highly conserved residues among the orthologous TNSALP proteins sequences from 15 different vertebrate species (Figure 2C). However, the alignment from protein sequences encoded by *ALPL*, *ALPI*, and *ALPP* revealed that only Ala33 residue is highly conserved among these paralogous human sequences (Figure 2D). With respect to their structure and functional impact, both genetic alterations were distant from the active site and would not be expected to directly affect the catalytic properties of TNSALP (Figure 2E). The 3D modeling analysis showed that the affected Ala33Val residue is located in the N-terminal α-helix, whereas the affected Asn47His residue is located in the coil structure close to N-terminal α-helix. Both p.Ala33Val and p.Asn47His residues are in close proximity to the homodimer interface.

Intramolecular and intermolecular contact analyses allowed additional predictions of the molecular effects of p.Asn47His and p.Ala33Val variants, showing that the replacement of Asn by His at position 47 may result in a gain of a hydrogen bond between the Asn47 and Thr48 residues (Figure 2F). However, no changes in the intramolecular contacts were predicted when Ala was replaced by Val at position 33 (Figure 2G).

## 4. Discussion

We describe a novel combination of biallelic *ALPL* missense variants in a female patient with childhood HPP, associated with skeletal mineralization defects including abnormal trabecular patterning at several sites, a fracture in the fibula, and craniosynostosis, as well as premature tooth loss and enamel defects. Computational modeling revealed the affected Asn47His residue is located in the coil structure close to N-terminal α-helix, whereas the affected Ala33Val residue is localized in the N-terminal α-helix, both located close to the homodimer interface, suggesting that both mutations might impair TNSALP dimer formation and stability. For six years, ERT was associated with improvements in growth, the skeleton, and the dentition.

Severe forms of HPP are rare, with an estimated prevalence of 1/300,000 in the European population, 1/450,000 in Japan, 1/100,000 in the Toronto region of Canada, and 1/2500 in Mennonite families in Manitoba, Canada [7]. The prevalence of milder forms of HPP is difficult to estimate, and individuals are frequently misdiagnosed in the absence of biochemical and genetic information [8].

While genotype–phenotype correlations have been established for some HPP-causing *ALPL* mutations [9], the majority are not well understood regarding how the amino acid location and type of mutations contribute to biochemical, musculoskeletal, and neurological effects. In addition, it remains to be elucidated whether the prognosis for ERT is differentially impacted by the nature of *ALPL* variant(s) in a patient. In the current study, two variants were identified in the proband, both located in exon 3 (NM000478.6; exon 2 in CCDS): p.Ala33Val and p.Asn47His. The p.Ala33Val variant (described as Ala16Val in the previous nomenclature) has been previously reported in the NCBI database of genetic variation (dbSNP # rs121918005) and observed in individuals affected with HPP [10,11,12,13,14]. The p.Asn47His variant has not been previously reported. This missense variant replaces asparagine (an aliphatic amino acid) with histidine (imidazole side chain, which is partially protonated, and a Brønsted acid and base), amino acids with distinct properties. Another variant affecting the same codon (NM_000478.6:c.140A > T, p.Asn47Ile) was recently reported in the ClinVar database (accession number: VCV001334791.1) in an individual with homozygous, lethal HPP, and also in a heterozygous individual with adult HPP [15]. This variant is currently classified as likely pathogenic in the curated *ALPL* gene variant database (https://alplmutationdatabase.jku.at/, accessed on 8 December 2022). Unfortunately, the proband’s maternal grandmother, who lost her teeth prematurely, was unavailable for molecular analysis. Therefore, we cannot exclude the possibility that she also harbored the variant identified in her daughter and grandchild (p.Ala33Val), showing a mild form of HPP.

Previous studies have reported the impact of *ALPL* missense variants on enzyme activity, dimer stability, allosteric proprieties, uncompetitive inhibition, and substrate specificity and on catalytic activity, cell localization, and degradation of the mutant proteins, contributing to greater understanding of genotype–phenotype relationships [2]. Residual activity of *ALPL* variants located in the active site or vicinity, crown domain, and homo-dimer interface were demonstrated to be more negatively affected than variants in other domains [16]. The p.Ala33Val variant was previously reported to have a very low residual enzyme activity compared with the wild-type protein [15,16]. Although bioinformatic approaches predicted both variants to be likely pathogenic and further computational analysis suggested the location of the affected residues could impact TNSALP dimer stability, additional functional studies should be performed in order to confirm these predictions. Biallelic, compound heterozygous *ALPL* variants are increasingly reported for HPP, including for the Ala33 position involved in this case. Different HPP subtypes are associated with the p.Ala33Val variant: p.[Ala33Val];[Tyr236Cys] and p.[Ala33Val];[Ala33Val] genotypes were associated with perinatal HPP [10,11], whereas p.[Ala16Val];[Tyr436His], p.[Ala16Val];[Pro275Thr], and p.[Ala33Val];[Glu191Lys] were associated with infantile and severe childhood HPP, respectively [12,13,14].

The proband was treated with asfotase alfa (Strensiq™; Alexion Pharmaceuticals) ERT starting at two years old. Asfotase alfa is a hydroxyapatite-targeted recombinant form of TNSALP indicated for use in life-threatening perinatal/infantile and severe childhood or juvenile-onset cases of HPP in the United States, Canada, Europe, and Japan [17]. Patients treated with asfotase alfa exhibited dramatic improvements in survival, bone mineralization, growth, respiratory function, and mobility [17,18]. After six years of ERT, the proband exhibited improvements in growth, motor skills, and radiological appearance. Although clinical improvements were observed after ERT, BMD analysis failed to show substantial changes overtime. Caution should be used to interpret these findings, as a recent study has suggested that DXA BMD Z-scores presented limited value to detect deficient bone mineralization in patients with HPP [19]. Very few studies have described effects of ERT on the dentition. Schroth et al. reported that early vs. late administration of ERT to children with infantile HPP resulted in reduced rates of premature tooth exfoliation [20]. Although ERT began at two years, the proband prematurely lost deciduous incisors, canines, and molars. However, at age eight, she presented with a normal and clinically stable permanent dentition, with the exception of mild enamel defects. Indirectly, these findings suggest ERT may have contributed to the stabilization of her remaining deciduous and permanent dentitions.

## 5. Conclusions

In conclusion, the novel combination of compound heterozygous recessive variants identified in this proband was associated with skeletal and dental manifestations of HPP, including decreased motor skills, short stature, craniosynostosis, low bone density, and premature deciduous tooth loss. ERT positively impacted skeletal and dental mineralization, leading to an overall clinical improvement.

## Figures and Tables

**Figure 1 ijms-24-00282-f001:**
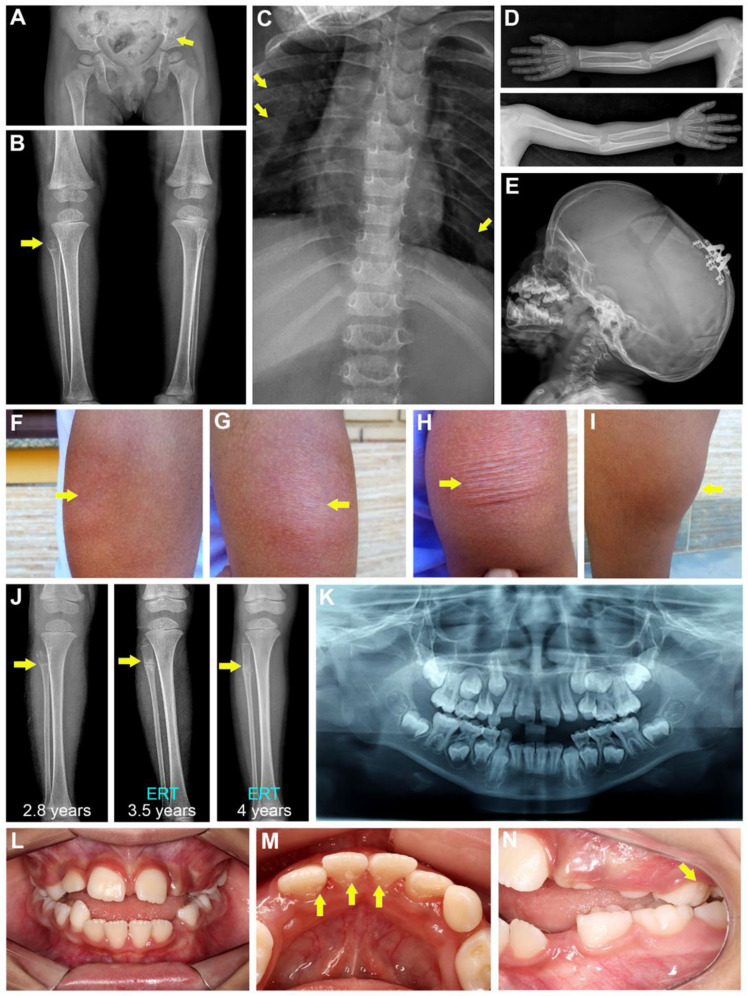
Clinical presentation. Representative radiographs of the proband at 2 years of age show (**A**,**B**) abnormal heterogeneous trabecular bone in the ilium (yellow arrow in **A**), proximal femur, and the long bones at the knees, and radiolucent areas in the proximal metaphysis of the fibula (yellow arrow in **B**) and in the central metaphysis and epiphysis of the left distal femur. (**C**) The proband shows enlargement of the anterior costochondral junction (yellow arrows). (**D**) No abnormalities were observed for the long bones of the upper limbs. (**E**) Note a dolichocephalic skull and the presence of distractors after cranial vault decompression. (**F**–**H**) Asfotase alfa ERT was associated with injection site skin reactions including redness, swelling, and thickening (yellow arrows indicated affected regions) and (**I**) lipodystrophy, as shown in this photograph of the proband’s thigh. (**J**) The radiolucency and traverse fracture detected in the proximal right fibula (white arrows) were resolved after introduction of asfotase alfa ERT, with follow-up images at 3 years and 5 months and 4 years. (**K**) Panoramic oral radiograph of the proband at 8 years of age shows a mixed dentition. (**L**–**N**) Intraoral photos at 8 years of age show mild enamel hypoplastic lesions on the lower permanent incisors and upper deciduous left molar (yellow arrows).

**Figure 2 ijms-24-00282-f002:**
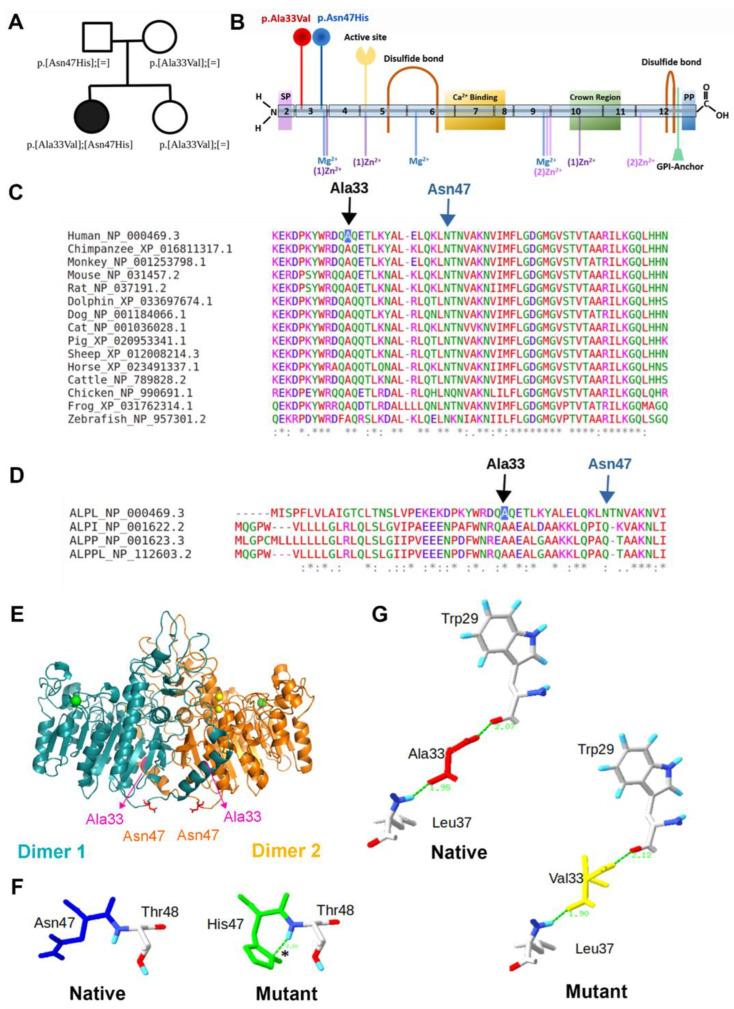
Genetic and pedigree analysis. (**A**) Pedigree chart representing two generations of the family analyzed in this study. Segregation patterns of genotypes (p.Ala33Val and/or p.Asn47His) and phenotype (serum ALP levels) for the proband (filled circle). (**B**) Protein model of TNSALP showing the signal peptide (SP) and functional regions, including active site, ion binding regions, crown domain, and disulfide bonds. Amino acid substitutions identified in the proband (p.Ala33Val and p.Asn47His) are indicated in exon 3 with red and blue markers. (**C**) Residue conservation of orthologous ALPL sequences across 15 vertebrate species. The Ala33 and Asn47 native residues are indicated by a black and a blue arrow, respectively. Both Ala33 and As47 residues (indicated by arrows) are highly conserved. (**D**) The residue conservation is displayed by multiple sequence alignment in paralogous sequences of human ALPL, ALPP, ALPI, and ALPPL genes. (**E**) 3D model of TNSALP dimer showing the localization of affected residues by p.Ala33Val and p.Asn47His mutations in monomer 1 (cyan) and monomer 2 (orange). The Ala33 residue is located in the N-terminal alpha helix (α-helix), and Asn 47 residue is located in the coil structure close to the N-terminal (α-helix). The Ala33 residue in the α-helix structure is colored in pink and indicated by arrows, while the Asn47 is highlighted in the stick model and colored in red. Mg ions are colored green, and Zn ions colored yellow. The images were generated using the PyMol software. (**F**) Intramolecular interactions established by native Ala33 and mutant Val33 residues. (**G**) Intramolecular interactions established by native Asn47 and mutant His47 residues. A predicted new hydrogen bond between His47 and Thr48 is indicated by an asterisk. Hydrogen bonds (dashed green lines) and their distances are indicated. The images were generated using the Swiss-PdbViewer v4.1.

## Data Availability

The data that support the findings of this study are available from the authors upon request.

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
