# Peer review of "Childhood Hypophosphatasia Associated with a Novel Biallelic ALPL Variant at the TNSALP Dimer Interface"

_ijms, 2022, doi:10.3390/ijms24010282_

Round 1
Reviewer 1 Report
The research presented by Martins et al. is very well written and nicely presented. It was very clear what was conducted and what the findings were for the proband. It was also clear that the use of ERT had beneficial outcomes for the affected individual. The only minor issue I have is with the font size for Figures 1B & 1E; it could be a slightly larger for the purposes of clarity.
Author Response
Reviewer 1:
“The research presented by Martins et al. is very well written and nicely presented. It was very clear what was conducted and what the findings were for the proband. It was also clear that the use of ERT had beneficial outcomes for the affected individual. The only minor issue I have is with the font size for Figures 1B & 1E; it could be a slightly larger for the purposes of clarity”.
Response: We thank this reviewer for highlighting the relevance of the manuscript. We believe the reviewer meant figures 2B and 2E instead of 1B and 1E as the latter do not have text. Font sizes were therefore revised in figures 2B and 2E as suggested.
Reviewer 2 Report
In this case report, the authors described a case of childhood hypophosphatasia with p.Ala33Val and p.Asn47His variants in the ALPL gene. The former is already listed in the database of ALPL mutations (https://alplmutationdatabase.jku.at/table/), while the latter is a novel one. Based on computational modeling, the authors suggested both affected residues are located close to the homodimer interface and speculated they may impair TNSALP dimer formation and stability. However, they have not provided any experimental data suggesting the impaired homodimerization. This reviewer thinks that the manuscript lacks scientific significance that may add new insight to the molecular pathogenesis of hypophosphatasia.
Author Response
Reviewer: 2
“In this case report, the authors described a case of childhood hypophosphatasia with p.Ala33Val and p.Asn47His variants in the ALPL gene. The former is already listed in the database of ALPL mutations (https://alplmutationdatabase.jku.at/table/), while the latter is a novel one. Based on computational modeling, the authors suggested both affected residues are located close to the homodimer interface and speculated they may impair TNSALP dimer formation and stability. However, they have not provided any experimental data suggesting the impaired homodimerization. This reviewer thinks that the manuscript lacks scientific significance that may add new insight to the molecular pathogenesis of hypophosphatasia”.
Response: Thank you very much for this comment. We agree with the reviewer that further functional studies would be helpful to confirm the “in silico” prediction. The manuscript has been revised in order to clarify this limitation (please see the Discussion section). We also summarize current knowledge about a p.Asn47Ile variant affecting the same ALPL codon, including references and the designation of “Likely pathogenic” from the curated ALPL gene variant database.
Reviewer 3 Report
The article describes a rare but serious disease. It is important to better understand this disease and provide knowledge to improve treatment and monitoring strategies. The presented case is very interesting and well documented, bringing new aspects of genetic conditions.
Question 1.
One of the symptoms/complications of hypophosphatasia is decreased bone mineral density. Was bone mineral density assessment performed in the presented patient? If yes, please describe at what age and what the results were. Please describe the method used to measure bone density.
Question 2.
Did the use of ERT affect bone mineral density?
Overall comment:
The topic is interesting and The article needs minor revisions as mentioned above.
Author Response
Reviewer: 3
“The article describes a rare but serious disease. It is important to better understand this disease and provide knowledge to improve treatment and monitoring strategies. The presented case is very interesting and well documented, bringing new aspects of genetic conditions”.
“Question 1”.
One of the symptoms/complications of hypophosphatasia is decreased bone mineral density. Was bone mineral density assessment performed in the presented patient? If yes, please describe at what age and what the results were. Please describe the method used to measure bone density.
“Question 2”.
Did the use of ERT affect bone mineral density?
Response: We appreciate your questions. Bone mineral density assessments were performed and are now described in the revised manuscript (please see the Results and Discussion sections).